# Ion exchange chromatography as a simple and scalable method to isolate biologically active small extracellular vesicles from conditioned media

Ricardo Malvicini[1,2,3,4]*, Diego Santa-Cruz[1], Anna Maria Tolomeo[3,4,5], Maurizio Muraca[2,3,4], Gustavo Yannarelli[1⊚], Natalia Pacienza[1⊚]

1 Laboratorio de Regulación Génica y Células Madre, Instituto de Medicina Traslacional, Trasplante y Bioingeniería (IMeTTyB), Universidad Favaloro-CONICET, Buenos Aires, Argentina, 2 Department of Women's and Children's Health, University of Padova, Padua, Italy, 3 Laboratory of Extracellular Vesicles as Therapeutic Tools, Fondazione Istituto di Ricerca Pediatrica Città della Speranza, Padua, Italy, 4 L.i.f.e.L.a.b. Program, Consorzio per la Ricerca Sanitaria (CORIS), Padua, Italy, 5 Department of Cardiac, Thoracic and Vascular Science and Public Health, University of Padova, Padua, Italy

⊚ These authors contributed equally to this work.
* rmalvicini@favaloro.edu.ar

## Abstract

In the last few years, extracellular vesicles (EVs) have become of great interest due to their potential as biomarkers, drug delivery systems, and, in particular, therapeutic agents. However, there is no consensus on which is the best way to isolate these EVs. The choice of the isolation method depends on the starting material (i.e., conditioned culture media, urine, serum, etc.) and their downstream applications. Even though there are numerous methods to isolate EVs, few are compatible with clinical applications as they are not scalable. In the present work, we set up a protocol to isolate EVs from conditioned media by ion exchange chromatography, a simple, fast, and scalable method, suitable for clinical production. We performed the isolation using an anion exchange resin (Q sepharose) and eluted the EVs using 500 mM NaCl. We characterized the elution profile by measuring protein and lipid concentration, and CD63 by ELISA. Moreover, we immunophenotyped all the eluted fractions, assessed the presence of TSG101, calnexin, and cytochrome C by western blot, analyzed nanoparticle size and distribution by tRPS, and morphology by TEM. Finally, we evaluated the immunomodulatory activity *in vitro*. We found that most EVs are eluted and concentrated in a single peak fraction, with a mean particle size of <150nm and expression of CD9, CD63, CD81, and TSG101 markers. Moreover, sEVs in fraction 4 exerted an anti-inflammatory activity on LPS-stimulated macrophages. In summary, we set up a chromatographic, scalable, and clinically compatible method to isolate and concentrate small EVs from conditioned media, which preserves the EVs biological activity.

**Data Availability Statement:** All relevant data are within the manuscript and its Supporting Information files.

**Funding:** This work was supported by the Fondo para la Investigación Científica y Tecnológica (FONCyT) under grants PICT-2019-00659 and PICT 2020-SERIE A-03292 (held by NP) and by Consejo Nacional de Investigaciones Científicas y Técnicas (CONICET) under grant PIP-2015-2017 (11220150100188CO) held by GY. This work was also supported by CONICET under a PUE grant (22920160100101CO) and by Consorzio per la Ricerca Sanitaria (LIFELAB Program) (grant no. DGR1017, July 17, 2018, held by MM), which also funded R. Malvicini air tickets from Argentina to Italy". The role of the founders should read as follows: "MM, GY and NP took part in the data curation, supervision and writing of the original manuscript and the revised version. GY and NP also took part in the decision to publish, conceptualization and data analysis.

**Competing interests:** The authors have declared that no competing interests exist.

## Introduction

In the last few years, extracellular vesicles smaller than 200 nm (sEVs) have drawn attention as they are implicated in numerous biological processes, including cell-to-cell communication, both in paracrine and endocrine fashions [1]. In particular, mesenchymal stromal cells-derived sEVs (MSC-sEVs) have been found to reproduce many of the beneficial biological activities of their parental cells, such as immunomodulatory, anti-inflammatory, antioxidant, and anti-apoptotic activities [2–5]. However, MSC-sEVs have several advantages compared to their parental cells, as they are non-living and non-replicating entities they do not have the capacity for ectopic colonization, and the risk of embolism after intravenous or intra-arterial administration is lower. Consequently, sEVs administration represents a safer and more attractive therapeutic approach [6–8]. In this sense, MSC-sEVs infusion has been used to treat different pathologies, such as acute myocardial infarction, acute kidney injury, inflammatory bowel disease, and autoimmune diseases, among others, in which the immune system plays a key role [5, 9–11]. sEVs are composed of a lipid bilayer with transmembrane proteins enclosing cytoplasmic components, resulting in complex biological entities that carry different molecules, including lipids, proteins, and nucleic acids (miRNA, mRNA, lncRNA, among many others) [12–14]. In this sense, it has been demonstrated that the cell secretome and the sEVs cargo are greatly influenced by the environment and the cell culture conditions [15]. Moreover, the characteristics and activity of a certain sEV preparation will also depend on the chosen isolation method, as it directly determines which EV subpopulations are isolated, and also, which other components are co-isolated (such as protein aggregates, RNAs, etc.) based on each method isolation principle [16–18]. Currently, several different methods are employed, including ultracentrifugation, density gradient centrifugation, size exclusion chromatography, polyethylene glycol precipitation, and tangential flow filtration (TFF). Until now, the most widely diffused methods to isolate EVs at lab-scale are ultracentrifugation and density gradients. However, these methods do not allow to process large volumes of culture media and, therefore, are unsuitable for the large-scale manufacturing required for clinical applications. In the sense, TFF, asymmetrical flow field-fractionation (AF4) and IEX are regarded as promising methods for the large-scale EV isolation, as they allow to process large volumes of conditioned media, are feasibly automated and are cost-effective [19, 20].

MSC-derived sEVs isolated by IEX have shown beneficial effects in animal models of traumatic brain injury, status epilepticus, LPS-induced systemic inflammation, arterial stiffness and hypertension, and allergic airway inflammation [21–25]. However, most of these reports do not provide a detailed isolation methodology. Moreover, none of them characterized the sEVs in compliance with ISEV criteria [26] and selected the sEV fraction based on protein content or CD63 expression.

In our current work, we optimized and fully characterized a previously described method to isolate MSC-derived sEVs from conditioned media by anion exchange chromatography using a Q Sepharose resin [23, 27]. As EVs have a negatively charged surface or zeta potential, they are first retained on the cationic resin by electrostatic forces and then eluted by increasing the ionic strength (salt concentration) of the buffer. This method is characterized for being soft (i.e., sEVs are not subjected to extreme centrifugal forces) and scalable allowing the processing of large amounts of conditioned media and making it compatible with clinical production. The performance of this isolation method was assessed: (I) by characterizing the biochemical profile of the eluted fractions; (II) by evaluating the presence of sEVs by transmission electron microscopy; (III) by characterizing the size and distribution by tRPS analysis; (IV) by detecting tetraspanins (CD9, CD63, and CD81) and the exosome marker TSG101; and, (V) by evaluating the biological activity of the eluted fractions. As a result, most of the

MSC-sEVs are eluted and concentrated in a single peak fraction, rendering a pure and biologically active sEV preparation.

## Materials and methods

The protocol described in this peer-reviewed article is published on protocols.io, dx.doi.org/10.17504/protocols.io.3byl4b71zvo5/v1 and is included for printing as S1 File with this article.

## Expected results and discussion

### MSC-sEVs isolation and elution profile characterization

Conditioned media (60-200mL) was subjected to ion exchange chromatography for EV isolation, and 8 fractions of 1 mL each were obtained (Fig 1). In order to characterize the elution profile and evaluate the presence of MSC-sEVs, we analyzed each fraction by tRPS and TEM. tRPS analysis was only possible in fractions 2 to 5, as in the other fractions no particles were detected (Fig 2A). The distribution parameters (mean, mode, D10, D50, and D90) of each fraction are summarized in the S1 Table. In all cases, the mean diameter was <200 nm, suggesting the presence of only sEVs. Moreover, we found that the particle number peaked at fraction 4 (Fig 2B).

However, when analysing the different fractions by TEM, vesicular structures were found in fractions 3 and 4, indicating that EVs are only present in these two fractions (Fig 3). To further characterize the elution profile, we evaluated the protein, lipid, and nucleic acids concentration in each fraction.

We found no proteins in fractions 1 and 2, and they begin to elute in fraction 3, peaking at fraction 5, and then steadily decreasing between fractions 6 to 8 (Fig 4A). Moreover, silver staining of the different fractions showed the same elution profile as BCA analysis, with proteins peaking in fraction 5, while the protein pattern revealed an enrichment of proteins bigger than 37KDa (Fig 4B). Moreover, almost a 50-fold protein concentration was achieved, when comparing fraction 4 to the conditioned media. Regarding the lipids, the highest amount of lipids was found in fraction 4. This may be due to the presence of EVs, as demonstrated by transmission electron microscopy imaging, which confirms the presence of vesicles only in fractions 3 and 4 (Fig 4C). Finally, we evaluated the expression of CD63 by a sandwich ELISA, which allows detecting particles with more than one molecule of CD63 (free CD63 is not

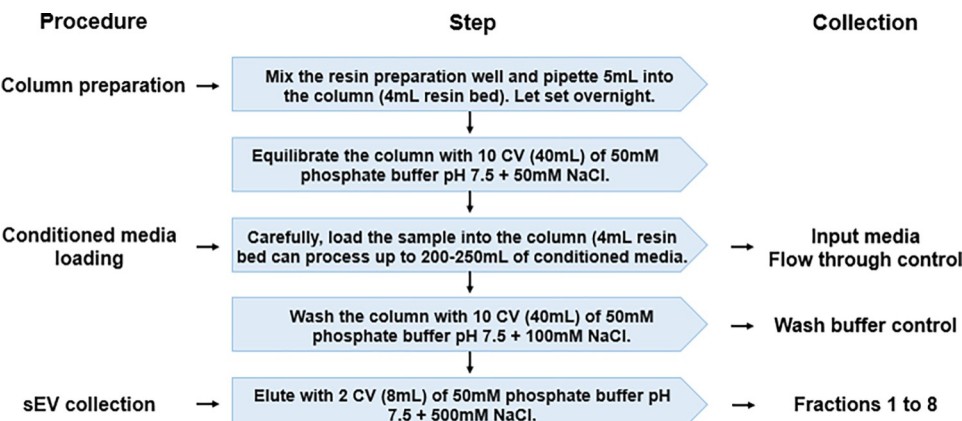

**Fig 1. Ion exchange chromatography protocol.** Schematic and summarized protocol for the isolation of sEVs from conditioned media by ion exchange chromatography.

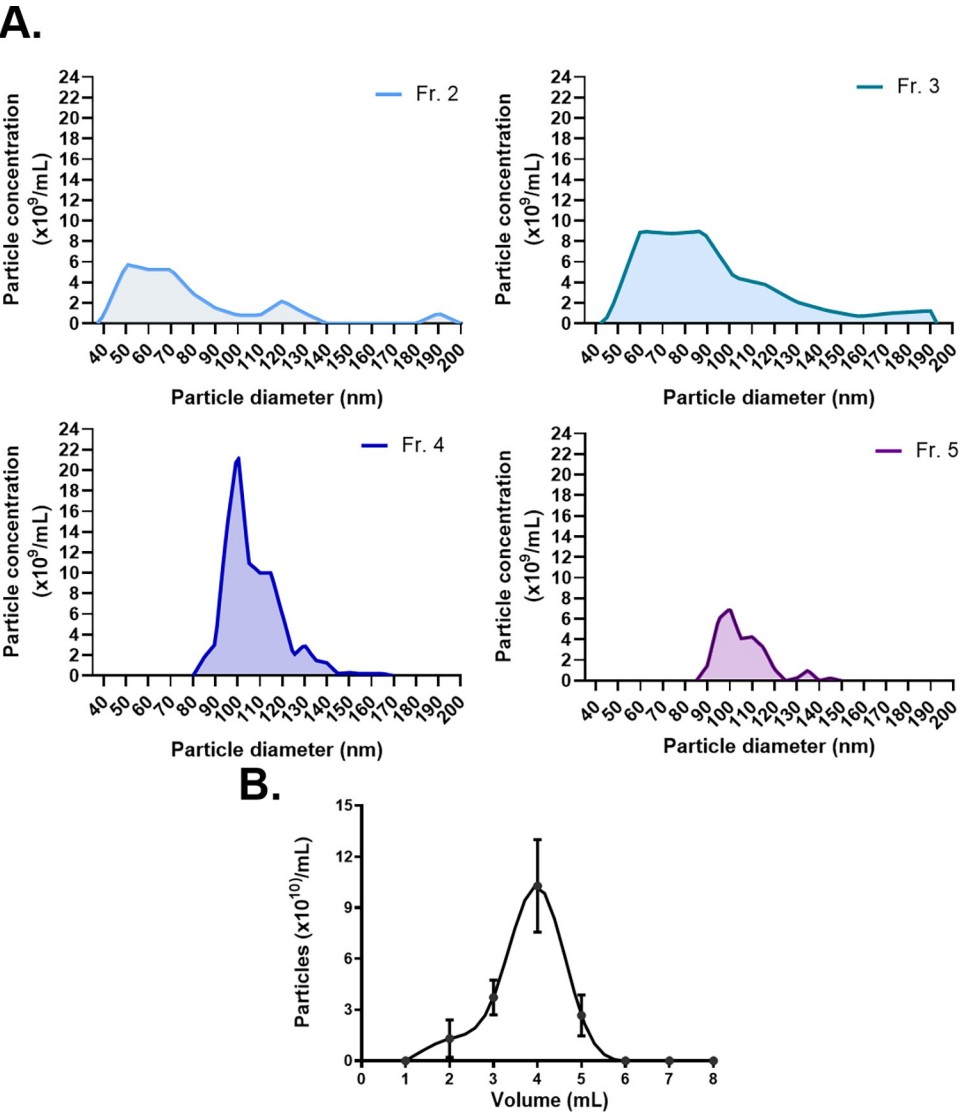

**Fig 2. Size and distribution analysis of the different fractions. (A)** Representative size distribution profile of fractions 2 to 5 measured by tunable resistive pulse sensing (tRPS). **(B)** Mean particle concentration for each fraction (mean±SD from five independent samples).

detected). We found that there was no CD63 in fractions 1 and 2, it then peaked in fraction 4, and decreased from fractions 5 to 8 (Fig 4D). These results show that, independently of the input volume, the method is robust and reproducible, as the same elution profile is observed (for all the isolations performed) for the proteins, lipids and CD63. As expected, the yield of proteins and lipids do vary according to the input volume. Finally, to evaluate whether the proteins were free or bound to the EVs, fraction 4 was dialyzed with a 300KDa dialysis membrane and then an SDS-PAGE followed by silver staining was performed. We found that the protein pattern before and after the dialysis is comparable, thus confirming that most of the proteins are bound to the EVs (Fig 4E).

We also evaluated the presence of the tetraspanins CD9, CD63, and CD81 in all eight fractions, by flow cytometry. Interestingly, all three tetraspanins showed a very similar elution profile among them and also with the CD63 elution profile evaluated by ELISA, as these markers

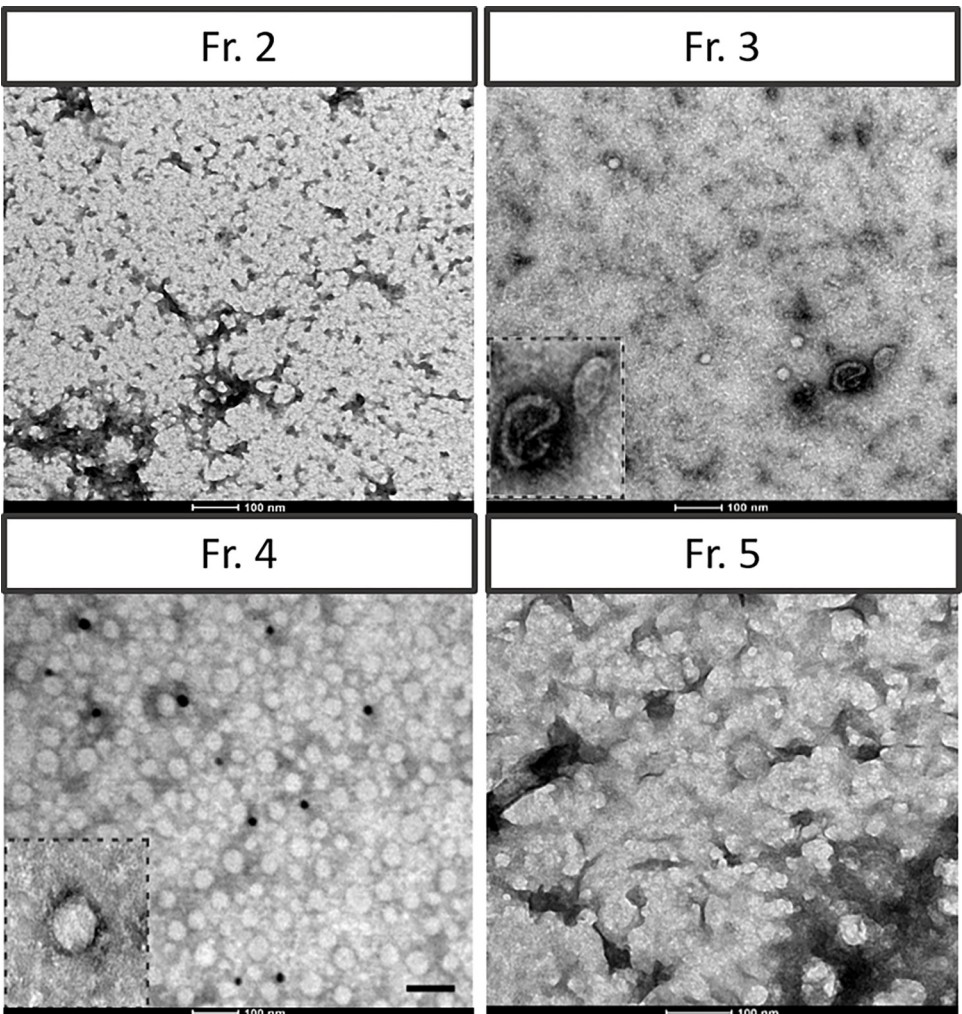

**Fig 3. Transmission electron microscopy (TEM) characterization.** Fractions 2 to 5, in which particles were detected, were subjected to TEM analysis. Vesicular structures were only found in fractions 3 and 4. Inset: close-up view of a vesicle.

were not detected in fractions 1 and 2, then peaked in fraction 4, and then decreased to almost undetectable levels from fraction 5 to 8. The major difference among the markers was that CD63 was the most abundant marker (almost 6-fold, with respect to CD9 and CD81) and it was also found in fraction 3, while CD9 and CD81 were not (Fig 5A). Finally, we evaluated the presence of the intravesicular exosome marker TSG101 in the different fractions. Remarkably, TSG101 elution profile corresponded to that of CD9, CD63, and CD81, as it was absent in fractions 1 and 2, peaked at fraction 4, decreased in fraction 5 and became undetectable in fractions 6 to 8 (Fig 5B). These results indicate that MSC-sEVs are mostly eluted as a single peak in fraction 4, with a minor fraction of them also eluted in fraction 3. Moreover, to rule out any possible contamination with mitochondria or endoplasmic reticulum derived vesicles, we evaluated the presence of cytochrome C and calnexin by western blot. We found that these proteins were present in MSCs, but were absent in all the eluted fractions (Fig 5C). The protein, lipid, and CD63 elution profiles from five different independent preparations, demonstrate that the method is robust and reproducible, even if processing different volumes of culture

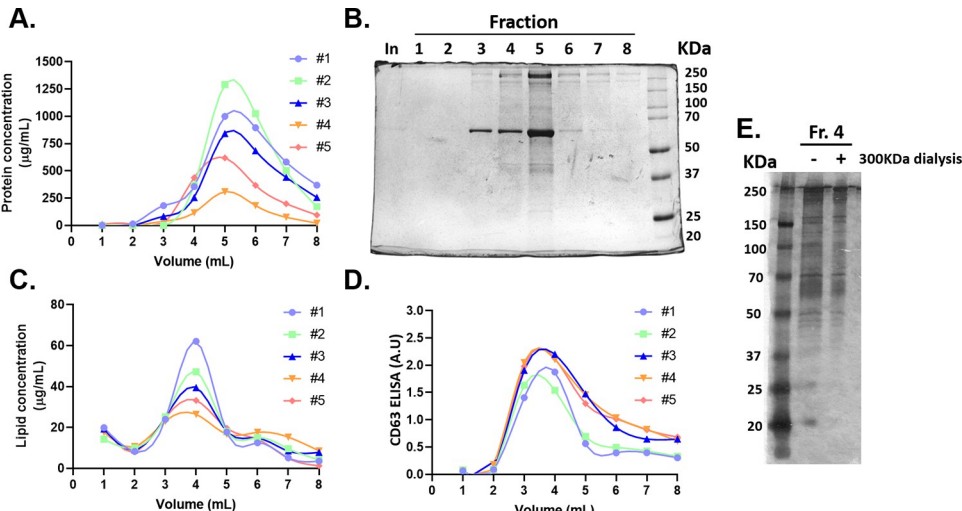

**Fig 4. Biochemical characterization of the eluted fractions.** Five independent EV isolations from conditioned media (volume ranging between 60-200mL) were performed. **(A)** Protein concentration was measured by BCA. **(B)** The protein pattern in the different fractions was evaluated by SDS-PAGE followed by silver staining. **(C)** Lipid concentration was assessed by sulfo-phospho-vanillin assay. **(D)** The presence of the exosome marker CD63 was assessed by ELISA. **(E)** The protein pattern in fraction 4 before and after a 300KDa dialysis was evaluated by SDS-PAGE followed by silver staining.

media. The protein recovery rate, calculated as: (fraction volume x fraction protein concentration) x100/(input volume x input protein concentration), is usually between 4–18% (mean ±2SD). It is important to consider that EVs are eluted in a high salt concentration buffer, which may affect downstream processing. As EVs are significantly concentrated during the isolation, it is often possible to dilute them with a buffer without NaCl to adjust the final salt concentration. Otherwise, EVs can be desalted by filtration using 100KDa Amicon filters.

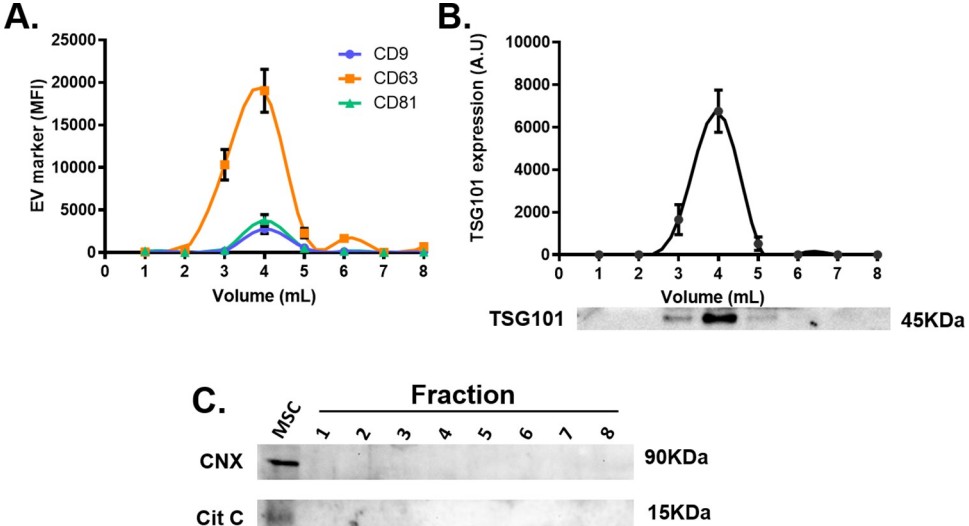

**Fig 5. Exosome markers analysis. (A)** The presence of the tetraspanins CD9, CD63, and CD81, which are considered exosome markers, were assessed by flow cytometry. **(B)** The intravesicular protein TSG101 was assessed by western blot and the optical density was quantified using Image J software. **(C)** The absence of cytochrome C and calnexin in all the eluted fractions was confirmed by western blot.

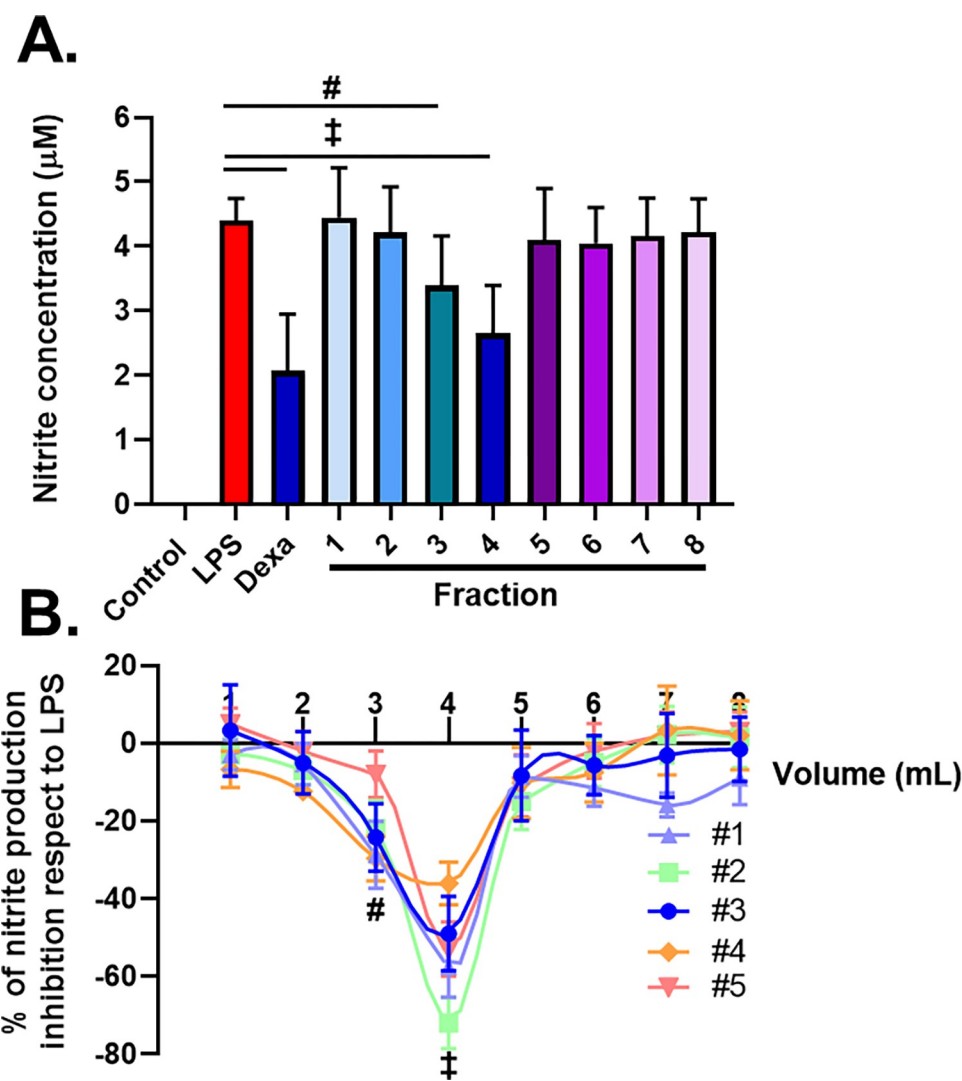

**Fig 6. Ion exchange chromatography yields biologically active sEVs.** The anti-inflammatory activity of the different fractions was tested on LPS-stimulated RAW264.7 macrophages. **(A)** Nitric oxide production was used as an M1 polarization index. In culture media, nitric oxide turns into nitrite, which was quantified by Griess reaction. Representative results from one independent isolation. **(B)** Activity of the different fractions calculated as the percentage of nitrite production inhibition with respect to LPS for each independent isolation. Results from three independent experiments. #$p < 0.01$; ‡$p < 0.001$ after One-Way ANOVA with Tukey's post test.

Additionally, we further characterized the different fractions by assessing the presence of MSC-related markers, adhesion molecules, and immunological markers by flow cytometry. Regarding MSC-related markers, we found that fraction 4 was positive for SSEA4 (detected also in fractions 3 and 5), CD44 (detected also in fractions 5 and 6) and CD105, while negative for HLA-1 (which was slightly positive in fraction 3), CD45 and HLA-DR (S1A Fig). Regarding the adhesion molecules, CD29 was highly expressed in fraction 4 (and was also detectable in fractions 3, 5 and 6), while CD41b, CD49e, CD62p, CD133, CD142 and CD146 were barely detectable (S1B Fig). Finally, immunological markers were almost not detectable in any fraction, except for low levels of CD3 (fractions 1, 2 and 3) and CD56 (all fractions except for fraction 4 and 6) (S1C Fig). In summary, surface protein expression in sEVs reflects that of the parent cells.

Finally, we evaluated the biological activity of the different eluted fractions for each independent isolation performed, to also test the reproducibility and robustness of the method. For this purpose, we employed a standardized in vitro assay that assesses EVs anti-inflammatory activity in LPS-stimulated macrophages [19, 20]. As shown in Fig 6A, macrophages significantly increased nitrite production after LPS stimulation, while the addition of dexamethasone significantly inhibited this response (-68 ± 19%; p<0.001). We found that fractions 3 and 4 exerted an anti-inflammatory activity, as they significantly inhibited nitrite production by about 10–30% and 35–75% (p<0.01 and p<0.001, respectively, respect to LPS), while the other fractions did not show any significant biological activity (Fig 6B). Even though the relative activity of fractions 3 and 4 varied between the replicates, the highest biological activity was always retrieved in fraction 4. Interestingly, the anti-inflammatory activity of the elution profile corresponds with the elution profile for CD63 and TSG101.

## Conclusion

In this work, we optimized and fully characterized a simple and soft method for sEV isolation using anion exchange chromatography, where sEVs are eluted with 500 mM NaCl. The elution profile of the EVs was characterized not only at protein and lipid level, but also by assessing the presence of specific markers, which indicate that EVs are concentrated and eluted mainly as a single fraction. At lab-scale, the ion exchange chromatography allows the isolation of sEVs from up to 200 ml of conditioned media, but as it is a scalable method, it would allow the isolation of sEVs from litres of conditioned media, compatible with clinical applications. Moreover, as it is a soft method and no extreme centrifugal forces are needed to isolate the EVs, only the gravity force, their morphology is preserved, contributing to their stability and retention of their biological activity, as demonstrated by the anti-inflammatory effect on LPS-stimulated macrophages. Our results suggest that EVs are mostly retrieved in fraction 4, in which the highest amount of CD9, CD63, CD81 and TSG101 are found as well as the highest biological activity is observed. Thus, it would be advisable to work only with fraction 4, without pooling the different eluted fractions. So far, the isolation methods regarded as compatible with the large-scale production of sEV are TFF, AF4 and IEX. In this regard, the present ion exchange chromatography protocol provides proof of concept of a methodology that is feasibly scalable and that allows the isolation and concentration of biologically active EVs in just 1mL. Finally, it should be useful for the isolation of sEV from large volumes of conditioned media, contributing to translating EV therapy from basic research into the clinic.

## Supporting information

**S1 File. Ion exchange chromatography protocol for the isolation of extracellular vesicles from conditioned media.** Step-by-step protocol, also available on protocols.io.
(PDF)

**S2 File. Materials and methods related to the production, quantification and characterization of the EVs.**
(DOCX)

**S3 File. Raw gel/blots images.** Uncropped and unadjusted images of the blots for TSG101, calnexin and Cytochrome C and the gels after silver-staining are provided.
(PDF)

**S4 File. MacsPlex flow cytometry gating strategy.** A representative gating strategy for the analysis of the different sample is provided.
(PDF)

**S5 File. MacsPlex flow cytometry raw data.** Representative RAW data (mean fluorescence intensity) from the MacsPlex analysis from a single experiment is provided.
(TXT)

**S1 Fig. sEV surface proteins assessment.** The presence of surface proteins was assessed by flow cytometry using the MACSPlex kit. MSC related proteins are depicted in (A), adhesion molecules are shown in (B) and immunological related proteins are shown in (C).
(DOCX)

**S1 Table. Size and distribution parameters.** Mean size, mode, D10, D50 and D90 parameters from the tRPS analysis for fractions 2, 3, 4 and 5.
(DOCX)

## Author Contributions

**Conceptualization:** Ricardo Malvicini, Diego Santa-Cruz, Maurizio Muraca, Gustavo Yannarelli, Natalia Pacienza.

**Data curation:** Ricardo Malvicini, Diego Santa-Cruz, Anna Maria Tolomeo.

**Formal analysis:** Ricardo Malvicini, Diego Santa-Cruz.

**Funding acquisition:** Maurizio Muraca, Gustavo Yannarelli, Natalia Pacienza.

**Investigation:** Ricardo Malvicini, Diego Santa-Cruz, Anna Maria Tolomeo.

**Methodology:** Ricardo Malvicini.

**Project administration:** Maurizio Muraca, Gustavo Yannarelli, Natalia Pacienza.

**Resources:** Maurizio Muraca, Gustavo Yannarelli.

**Supervision:** Maurizio Muraca, Gustavo Yannarelli, Natalia Pacienza.

**Validation:** Gustavo Yannarelli.

**Visualization:** Ricardo Malvicini.

**Writing – original draft:** Ricardo Malvicini, Diego Santa-Cruz, Maurizio Muraca, Gustavo Yannarelli, Natalia Pacienza.

**Writing – review & editing:** Ricardo Malvicini, Diego Santa-Cruz, Gustavo Yannarelli, Natalia Pacienza.

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
