## [Decision Letter · Decision Letter 0]

13 Jul 2023

PONE-D-23-17535Ion exchange chromatography as a simple and scalable method to isolate biologically active small extracellular vesicles from conditioned mediaPLOS ONE

Dear Dr. Malvicini,

Thank you for submitting your manuscript to PLOS ONE. After careful consideration, we feel that it has merit but does not fully meet PLOS ONE’s publication criteria as it currently stands. Therefore, we invite you to submit a revised version of the manuscript that addresses the points raised during the review process.

We look forward to receiving your revised manuscript.

Kind regards,

Elia Bari

Academic Editor

PLOS ONE

Comments from PLOS Editorial Office: We note that one or more reviewers has recommended that you cite specific previously published works. As always, we recommend that you please review and evaluate the requested works to determine whether they are relevant and should be cited. It is not a requirement to cite these works. We appreciate your attention to this request.

“This work was supported by the Fondo para la Investigación Científica y Tecnológica (FONCyT) under grants PICT-2019-00659 and PICT 2020-SERIE A-03292 (held by NP) and by Consejo Nacional de Investigaciones Científicas y Técnicas (CONICET) under grant PIP-2015-2017 (11220150100188CO) held by GY. This work was also supported by CONICET under a PUE grant (22920160100101CO) and by Consorzio per la Ricerca Sanitaria (LIFELAB Program) (grant no. DGR1017, July 17, 2018).”

“We are grateful to L.i.f.e.L.a.b. Program, Consorzio per la Ricerca Sanitaria (CORIS) for funding R. Malvicini air tickets from Argentina to Italy.”

“This work was supported by the Fondo para la Investigación Científica y Tecnológica (FONCyT) under grants PICT-2019-00659 and PICT 2020-SERIE A-03292 (held by NP) and by Consejo Nacional de Investigaciones Científicas y Técnicas (CONICET) under grant PIP-2015-2017 (11220150100188CO) held by GY. This work was also supported by CONICET under a PUE grant (22920160100101CO) and by Consorzio per la Ricerca Sanitaria (LIFELAB Program) (grant no. DGR1017, July 17, 2018).”

7. We note you have not yet provided a protocols.io PDF version of your protocol and/or a protocols.io DOI. When you submit your revision, please provide a PDF version of your protocol as generated by protocols.io (the file will have the protocols.io logo in the upper right corner of the first page) as a Supporting Information file. The filename should be S1_file.pdf, and you should enter “S1 File” into the Description field. Any additional protocols should be numbered S2, S3, and so on. Please also follow the instructions for Supporting Information captions [https://journals.plos.org/plosone/s/supporting-information#loc-captions]. The title in the caption should read: “Step-by-step protocol, also available on protocols.io.”

Please assign your protocol a protocols.io DOI, if you have not already done so, and include the following line in the Materials and Methods section of your manuscript: “The protocol described in this peer-reviewed article is published on protocols.io (https://dx.doi.org/10.17504/protocols.io.[...]) and is included for printing purposes as S1 File.” You should also supply the DOI in the Protocols.io DOI field of the submission form when you submit your revision.

If you have not yet uploaded your protocol to protocols.io, you are invited to use the platform’s protocol entry service [https://www.protocols.io/we-enter-protocols] for doing so, at no charge. Through this service, the team at protocols.io will enter your protocol for you and format it in a way that takes advantage of the platform’s features. When submitting your protocol to the protocol entry service please include the customer code PLOS2022 in the Note field and indicate that your protocol is associated with a PLOS ONE Lab Protocol Submission. You should also include the title and manuscript number of your PLOS ONE submission.

Reviewers' comments:

Reviewer's Responses to Questions

**Comments to the Author**

1. Does the manuscript report a protocol which is of utility to the research community and adds value to the published literature?

Reviewer #1: No

Reviewer #2: Yes

2. Has the protocol been described in sufficient detail?

To answer this question, please click the link to protocols.io in the Materials and Methods section of the manuscript (if a link has been provided) or consult the step-by-step protocol in the Supporting Information files.

The step-by-step protocol should contain sufficient detail for another researcher to be able to reproduce all experiments and analyses.

Reviewer #1: Partly

Reviewer #2: No

3. Does the protocol describe a validated method?

Reviewer #1: No

Reviewer #2: No

4. If the manuscript contains new data, have the authors made this data fully available?

Reviewer #1: Yes

Reviewer #2: Yes

**5. Is the article presented in an intelligible fashion and written in standard English?**

Reviewer #1: Yes

Reviewer #2: Yes

6. Review Comments to the Author

Reviewer #1: The manuscript (PONE-D-23-17535) entitled "Ion exchange chromatography as a simple and scalable method to isolate biologically active small extracellular vesicles from conditioned media" reports a method for isolating small extracellular vesicles (sEVs) from conditioned media utilizing anion exchange chromatography (AEX). The isolated sEVs were comprehensively characterized using different techniques. The manuscript is clearly written and easy to follow. However, given that this research focuses on the sEV isolation method, the study fails to provide any new or innovative approaches in the field, as the use of AEX with very similar protocol is already well-established for conditioned media (see Comment-2 in more detail). This has left no significant contribution or innovation from this study. Therefore, based on these grounds, it is recommended that the manuscript be rejected. Detailed comments are as follows.

Comment-1. Introduction section

In the introduction section (page 3-4), the authors discussed general isolation techniques of EVs and addressed the anion exchange chromatography (AEX) as a scalable and gentle method for EV isolation. The authors discussed that the mentioned EV isolation methods do not allow processing large volumes of culture media. This statement is not entirely true. For instance, tangential flow filtration (TFF) has been reported in several publications to isolate EVs from cell culture media. TFF does not require any chemical interaction unlike AEX and is also considered a gentle method for EV isolation due to the lack of chromatographic resins (similar to AF4). Pros and cons of each technique have been comprehensively reviewed in e.g. T. Liangsupree et al. (2021). Modern isolation and separation techniques for extracellular vesicles. Journal of Chromatography A, 1636, 461773.

To conclude, given that the highlights of this manuscript revolve around AEX being a scalable and gentle method for sEV isolation, the argument in the introduction is not strong enough to support why other techniques cannot be used instead of AEX. For instance, TFF and AF4 are also considered as soft techniques and automation is possible (unlike preparation AEX reported in this manuscript) and both techniques allow large scale separation and purification of EVs.

Comment-2. Expected Results and Discussion section

An almost identical protocol for isolation of biologically active sEVs from mesenchymal stromal cells has already been published elsewhere in 2020 (see below). Thus, as previously mentioned, as the main focus of this manuscript is sEV isolation, its novelty is compromised.

S. B. Fang et al. (2020). Small extracellular vesicles derived from human mesenchymal stromal cells prevent group 2 innate lymphoid cell-dominant allergic airway inflammation through delivery of miR-146a-5p. Journal of extracellular vesicles, 9(1), 1723260.

General comments to the method are given below:

1. Experiment details on MSC-sEVs isolation and elution profile characterization should (Page 5, MSC-sEVs isolation and elution profile characterization) should be part of Materials and Methods section and not in Expected Results and Discussion section.

2. It is reported that the column can be used for 200 ml of conditioned media. Extra information that could affect separation and yield, such as number of cells in the conditioned media, the times one prepared column can be used with satisfactory results (i.e., without losing its efficiency), should be given to claim that the method is reproducible especially when the column is manually prepared/packed.

3.What is the superiority of this method over other reported AEX methods also used for MSC derived EV isolation? The method reported in manuscript is laborious and requires over 4 hours to isolate EVs from 200 ml of condition media. Despite the fact that it is significantly shorter than ultracentrifugation, an AEX method that can process 1 L of conditioned media in less than 3 h has been reported (see N. Heath et al., (2018). Rapid isolation and enrichment of extracellular vesicle preparations using anion exchange chromatography. Scientific reports, 8(1), 5730).

A strong argumentative discussion is required to support the method reported in this manuscript.

4. The study reports the results from five independent samples, but no information on column-to-column reproducibility is provided.

Reviewer #2: The manuscript submitted by Ricardo Malvicini and the co-authors is devoted to the use of anion-exchange isolation of extracellular vesicles on Q-Sepharose.

The method proposed by the authors let them isolate the subfraction of sEV from a cell culture medium using a single chromatography step. The protocol is submitted to protocols.io, making it easily accessible to the readers.

The manuscript has several major and minor issues, which should be resolved before the manuscript might be accepted for publication.

1) The authors used the tRPS method for the sEV analysis. tRPS is first mentioned on p.5 of the pdf but is never deciphered. Also, the advantages and, disadvantages, limitations of the method are not discussed. This approach is not one of the most widely used for the characterization of sEV since this is obligatory for the manuscript

2) What is the protein with molecular mass between 50 and 70 kDa (Fig.4B, line 5)? It can be easily detected by trypsinolysis and MALDI of another mass-spectrometry approach. Whether this is serum albumin? If it is, whether these fractions contain sEV or are co-isolated with albumin during AEC on Q-Sepharose? The protein with ~250 kDa on line 5 can also be easily detected with any MS method.

Since the proteins in isolated fractions are easily distinguished, the basic proteomic investigation of fractions eluted from Q-Sepharose is highly recommended.

3) The authors state that "the highest amount of lipids was found in fraction 4, which corresponds to the presence of EVs," but this statement never provides evidence of EV-nature of this fraction.

4) The raw data of flow cytometry should be provided in the paper or the supplementary file

5) In the current form of the paper, the statement "the present ion exchange chromatography provides an alternative methodology that should be useful for the isolation of EVs" is not confirmed by experimental results.

Sincerely

7. PLOS authors have the option to publish the peer review history of their article (what does this mean?). If published, this will include your full peer review and any attached files.

Reviewer #1: No

Reviewer #2: No

---

## [Author Response · Author response to Decision Letter 0]

19 Aug 2023

Reviewer #1: 

The manuscript (PONE-D-23-17535) entitled "Ion exchange chromatography as a simple and scalable method to isolate biologically active small extracellular vesicles from conditioned media" reports a method for isolating small extracellular vesicles (sEVs) from conditioned media utilizing anion exchange chromatography (AEX). The isolated sEVs were comprehensively characterized using different techniques. The manuscript is clearly written and easy to follow. However, given that this research focuses on the sEV isolation method, the study fails to provide any new or innovative approaches in the field, as the use of AEX with very similar protocol is already well-established for conditioned media (see Comment-2 in more detail). This has left no significant contribution or innovation from this study. Therefore, based on these grounds, it is recommended that the manuscript be rejected. Detailed comments are as follows.

Comment-1. Introduction section

In the introduction section (page 3-4), the authors discussed general isolation techniques of EVs and addressed the anion exchange chromatography (AEX) as a scalable and gentle method for EV isolation. The authors discussed that the mentioned EV isolation methods do not allow processing large volumes of culture media. This statement is not entirely true. For instance, tangential flow filtration (TFF) has been reported in several publications to isolate EVs from cell culture media. TFF does not require any chemical interaction unlike AEX and is also considered a gentle method for EV isolation due to the lack of chromatographic resins (similar to AF4). Pros and cons of each technique have been comprehensively reviewed in e.g. T. Liangsupree et al. (2021). Modern isolation and separation techniques for extracellular vesicles. Journal of Chromatography A, 1636, 461773.

To conclude, given that the highlights of this manuscript revolve around AEX being a scalable and gentle method for sEV isolation, the argument in the introduction is not strong enough to support why other techniques cannot be used instead of AEX. For instance, TFF and AF4 are also considered as soft techniques and automation is possible (unlike preparation AEX reported in this manuscript) and both techniques allow large scale separation and purification of EVs.

Answer: We thank the reviewer for his/her comments and agree with him/her. We aim to propose the anion exchange chromatography as an alternative method for the large-scale production of sEV to TFF and AF4, but these techniques are also valid as scalable isolation methods. We realized that we failed to conveyed this idea, so we have changed the introduction section accordingly, as follows: “In the sense, TFF, asymmetrical flow field-fractionation (AF4) and IEX are regarded as promising methods for the large-scale EV isolation, as they allow to process large volumes of conditioned media, are feasibly automated and are cost-effective (Monguió-Tortajada et al.; Staubach et al.)”

Comment-2. Expected Results and Discussion section

An almost identical protocol for isolation of biologically active sEVs from mesenchymal stromal cells has already been published elsewhere in 2020 (see below). Thus, as previously mentioned, as the main focus of this manuscript is sEV isolation, its novelty is compromised.

S. B. Fang et al. (2020). Small extracellular vesicles derived from human mesenchymal stromal cells prevent group 2 innate lymphoid cell-dominant allergic airway inflammation through delivery of miR-146a-5p. Journal of extracellular vesicles, 9(1), 1723260 

Answer: We thank the reviewer for his/her comment. As we stated in the introduction section, we do not claim that this is a brand new protocol, but an optimized and fully characterized version of it. This protocol was used to successfully isolate EVs that were utilized in different animal models (Kim et al.; Pacienza et al.; Guan et al.; Feng et al.; Longa et al.; Fang et al.). However, to our knowledge, the methodology has not been fully characterized so far, as most works only provide a deficient characterization and only address protein concentration and CD63 expression in the fractions. Instead, in this work, and following the recommendations of the International Society for Extracellular Vesicles (ISEV), we performed an exhaustive characterization of the method and all the eluted fractions, by assessing the particle size and distribution (tRPS), morphology (TEM), protein and lipid quantification, EV positive (CD63, CD9, CD81, TSG101) and negative (Cytochrome C and calnexin) markers, EV surface markers and their biological activity. Moreover, we are providing a fully-detailed protocol, while in other works, only the general considerations are provided. 

General comments to the method are given below:

1. Experiment details on MSC-sEVs isolation and elution profile characterization should (Page 5, MSC-sEVs isolation and elution profile characterization) should be part of Materials and Methods section and not in Expected Results and Discussion section.

Answer: We thank the reviewer for his/her comment, and we removed that paragraph from the Expected Results and Discussion section.

2. It is reported that the column can be used for 200 ml of conditioned media. Extra information that could affect separation and yield, such as number of cells in the conditioned media, the times one prepared column can be used with satisfactory results (i.e., without losing its efficiency), should be given to claim that the method is reproducible especially when the column is manually prepared/packed.

Answer: We thank the reviewer for his/her comment. We believe that the maximum input volume that a 4mL resin can process will vary depending on the input media. At the same time, the input media will vary depending on the culture conditions (time, stimulation, etc), the cell type cultured, the cellular density and other factors. In this sense, we performed the isolation of EVs derived from MSCs after 48h of culture in alpha-mem, without FBS. At the time of medium change to alpha-MEM, the cellular confluence was about 70% and at the end of the culture, we retrieved about 3x106 cells/T175cm2 flask. 

In the different isolations, we processed between 60-200mL of conditioned media, that would be, the vesicles coming from 9-30x106 MSC. The results and the reproducibility of isolating different volumes of conditioned media can be seen in Figure 4A, C and D, where it can be observed the same elution profile, regardless of the input volume, but the yield varies according to the input volume. Moreover, we have also modified Figure 6 B, in which we show the anti-inflammatory activity of all the fractions, for each independent isolation performed, which also supports the reproducibility of the method.

To our knowledge and based on our results, the resin can be regenerated and reused up to two times, without losing its efficiency and maintaining the elution profile. Further re-utilizations would require validation.

We added the following paragraph in the supporting information file 2, MSCs culture section: “We performed 5 independent isolations and collected between 60-200mL of conditioned media, to evaluate the robustness and the reproducibility of the isolation protocol. The conditioned media collected contained the sEV produced by approximately 9-30x106 MSC or 3-10 T175 cm2 flasks”. 

Also, we added the following sentences in the results section “These results show that, independently of the input volume, the method is robust and reproducible, as the same elution profile is observed (for all the isolations performed) for the proteins, lipids and CD63. As expected, the yield of proteins and lipids do vary according to the input volume.” And “We found that fractions 3 and 4 exerted an anti-inflammatory activity, as they significantly inhibited nitrite production by about 10-30% and 35-75% (p<0.01 and p<0.001, respectively, respect to LPS), while the other fractions did not show any significant biological activity (Fig 6B). Even though the relative activity of fractions 3 and 4 varied between the replicates, the highest biological activity was always retrieved in fraction 4.”

Moreover, we added the following sentence in the Figure’s 4 figure legend: “Five independent EV isolations from conditioned media (volume ranging between 60-200mL) were performed.”

3.What is the superiority of this method over other reported AEX methods also used for MSC derived EV isolation? The method reported in manuscript is laborious and requires over 4 hours to isolate EVs from 200 ml of condition media. Despite the fact that it is significantly shorter than ultracentrifugation, an AEX method that can process 1 L of conditioned media in less than 3 h has been reported (see N. Heath et al., (2018). Rapid isolation and enrichment of extracellular vesicle preparations using anion exchange chromatography. Scientific reports, 8(1), 5730).

A strong argumentative discussion is required to support the method reported in this manuscript.

Answer: The method described in this manuscript is intended for the isolation of low to middle volumes of culture media, in a research laboratory setting. If more than 200 mL of conditioned media need to be processed, multiple columns can be run in parallel, in order to speed up the isolation process, and the final fractions can be pooled. We agree with the reviewer that if the intention is to isolate sEV from large volumes of culture media (i.e. 1 liter or more) the protocol we propose should be adapted (for instance, by using a larger resin volume, by using a peristaltic pump, etc) for this aim. Nonetheless, the method we propose is not laborious, as it only requires to load the buffers and the conditioned media to the column, and then wait for it to elute.

Regarding the manuscript by Heath et al. the reviewer mentioned, Heath et al. claim that they isolate EVs from 1 L of conditioned media coming from an average of 7.8x108 HEK293T cells, which were cultured using Corning® 10 layer CellSTACKs® (Sigma) and they obtain 2.4x1011 particles. However, in our experimental settings, we are isolating EVs from 0.2 L of conditioned media, coming from an average of 3.5x107 mesenchymal stromal cells, using Corning T175 flasks, and we obtain a mean value of about 1x1011 particles. Moreover, HEK293T cells are an immortalized cell line, and it is well documented in literature that cell lines and tumor-derived cells produce more EVs than primary cells (Whiteside; Chin and Wang; Logozzi et al.). In this sense, it would be expected to isolate a higher amount of EVs from HEK293T cells than from mesenchymal stromal cells. However, in our conditions, we isolated almost half the EVs Heath et al. isolated (1x1011 particles vs 2.4x1011 particles), from 20 times less cells (3.5x107 vs 7.8x108 ) and 5 times less conditioned media (0.2L vs 1L). These results would suggest the superiority of our method in the efficiency to isolate extracellular vesicles from conditioned media. Moreover, the isolation method we propose would render the EV production process less expensive, than that proposed by Heath et al., as to obtain the same yield of EVs we need to process 2.5 times less conditioned media. From this analysis, it would seem that despite the fact that the protocol proposed by Heath et al. allows to process large volumes of conditioned media, it presents a low recovery and a low yield, compared to the protocol we propose. Finally, our protocol establishes the elution of EVs with 500mM of NaCl, while that of Heath et al. performs the elution with 890mM of NaCl. In this sense, the higher salt concentration and ionic force may compromise the integrity of the EVs and their biological activity, and therefore, limit their application. In this work, we demonstrate that EVs eluted with 500mM of NaCl retain their biological activity, as they are able to inhibit nitrite production in LPS-stimulated macrophages.

Despite all these considerations, in order to claim that an isolation protocol is superior and yields more vesicles than another protocol, the same culture conditions and the same cell type should be employed, so that the input media is the same.

4. The study reports the results from five independent samples, but no information on column-to-column reproducibility is provided.

Answer: The five independent samples are depicted in Figure 4, A, C and D, which show that the same elution profile is maintained regardless of the input volume and thus, confirming the reproducibility of the method. Moreover, we have also modified Figure 6 B, in which we show the anti-inflammatory activity of all the fractions, for each independent isolation performed, which provides evidence of the reproducibility of the method.

Reviewer #2:

 The manuscript submitted by Ricardo Malvicini and the co-authors is devoted to the use of anion-exchange isolation of extracellular vesicles on Q-Sepharose.

The method proposed by the authors let them isolate the subfraction of sEV from a cell culture medium using a single chromatography step. The protocol is submitted to protocols.io, making it easily accessible to the readers.

The manuscript has several major and minor issues, which should be resolved before the manuscript might be accepted for publication.

1) The authors used the tRPS method for the sEV analysis. tRPS is first mentioned on p.5 of the pdf but is never deciphered. Also, the advantages and, disadvantages, limitations of the method are not discussed. This approach is not one of the most widely used for the characterization of sEV since this is obligatory for the manuscript

Answer: We thank the author for his/her comments. We clarified that tRPS stands for tunable resistive pulse sensing. Even though tRPS may not be the most diffused method to analyze particles’ size and distribution, it is recommended as a valid method of analysis by the international society for extracellular vesicles (ISEV), along with the Nanoparticle Tracking Analysis (NTA) and dynamic light scattering (DLS) (Théry et al.). Although it has some disadvantages, in our hands, this quantification method performs well. It is true that the quantification with the NTA gives a higher dynamic range than the tRPS, due to the fact that the membrane pore size limits the size of particles that can be quantified. However, as we filter the conditioned media by 0.22μM, we do not expect vesicles larger than this. Moreover, as it can be seen in Figure 2A, the whole peak is seen in fraction 4 and no particles bigger than 150nm are detected. The NP150 membrane that we used has an analysis range of 50-420nm, according to the manufacturers’ instructions.

We believe that discussion of the advantages, disadvantages and limitations of this method of analysis is out of the scope of this manuscript, as we are focusing on the isolation protocol. 

2) What is the protein with molecular mass between 50 and 70 kDa (Fig.4B, line 5)? It can be easily detected by trypsinolysis and MALDI of another mass-spectrometry approach. Whether this is serum albumin? If it is, whether these fractions contain sEV or are co-isolated with albumin during AEC on Q-Sepharose? The protein with ~250 kDa on line 5 can also be easily detected with any MS method.

Since the proteins in isolated fractions are easily distinguished, the basic proteomic investigation of fractions eluted from Q-Sepharose is highly recommended.

Answer: We thank the reviewer for his/her comments. For EV production, MSCs are expanded in DMEM supplemented with FBS. When they reach 70% confluence, cells are subjected to 3 washes with PBS, in order to eliminate all the serum, and then the media is changed to alpha-MEM (chemically defined media with amino acids and no proteins) without FBS. Therefore, albumin contamination would be unlikely. Moreover, we have performed the proteomic analysis of fraction 4, but we focused on the whole preparation of sEV and not in specific bands. Of note, albumin was not detected in our preparations after the proteomic analysis.

3) The authors state that "the highest amount of lipids was found in fraction 4, which corresponds to the presence of EVs," but this statement never provides evidence of EV-nature of this fraction.

Answer: We thank the reviewer for this comment and we modified this sentence in the manuscript as follows “the highest amount of lipids was found in fraction 4. This may be due to the presence of EVs, as demonstrated by transmission electron microscopy imaging, which confirms the presence of vesicles only in fractions 3 and 4”.

4) The raw data of flow cytometry should be provided in the paper or the supplementary file.

Answer: We provide the both the dot plots, showing the gating strategy, and the RAW data of one representative experiment as Supplementary file 4 (S4 MacsPlex gating strategy) and Supplementary file 5 (S5 MacsPlex Raw Data).

5) In the current form of the paper, the statement "the present ion exchange chromatography provides an alternative methodology that should be useful for the isolation of EVs" is not confirmed by experimental results.

Answer: We modified this sentence in the manuscript as follows: “So far, the isolation methods regarded as compatible with the large-scale production of sEV are TFF, AF4 and IEX. In this regard, the present ion exchange chromatography protocol provides proof of concept of a methodology that is feasibly scalable and that allows the isolation and concentration of biologically active EVs in just 1mL. It should be useful for the isolation of sEV from large volumes of conditioned media, contributing to translate EV therapy from basic research into the clinics.”

Bibliography

Chin, Andrew R., and Shizhen Emily Wang. “Cancer-Derived Extracellular Vesicles: The ‘Soil Conditioner’ in Breast Cancer Metastasis?” Cancer Metastasis Reviews, vol. 35, no. 4, NIH Public Access, Dec. 2016, p. 669, doi:10.1007/S10555-016-9639-8.

Fang, Shu Bin, et al. “Small Extracellular Vesicles Derived from Human Mesenchymal Stromal Cells Prevent Group 2 Innate Lymphoid Cell-Dominant Allergic Airway Inflammation through Delivery of MiR-146a-5p.” Journal of Extracellular Vesicles, vol. 9, no. 1, J Extracell Vesicles, Jan. 2020, doi:10.1080/20013078.2020.1723260.

Feng, Rui, et al. “Stem Cell-Derived Extracellular Vesicles Mitigate Ageing-Associated Arterial Stiffness and Hypertension.” Journal of Extracellular Vesicles, vol. 9, no. 1, J Extracell Vesicles, Jan. 2020, doi:10.1080/20013078.2020.1783869.

Guan, Xiaohui, et al. “MiR-223 Regulates Adipogenic and Osteogenic Differentiation of Mesenchymal Stem Cells Through a C/EBPs/MiR-223/FGFR2 Regulatory Feedback Loop.” STEM CELLS, vol. 33, no. 5, Wiley-Blackwell, May 2015, pp. 1589–600, doi:10.1002/stem.1947.

Kim, Dong Ki, et al. “Chromatographically Isolated CD63+CD81+ Extracellular Vesicles from Mesenchymal Stromal Cells Rescue Cognitive Impairments after TBI.” Proceedings of the National Academy of Sciences of the United States of America, vol. 113, no. 1, National Academy of Sciences, Jan. 2016, pp. 170–75, doi:10.1073/PNAS.1522297113/SUPPL_FILE/PNAS.201522297SI.PDF.

Logozzi, Mariantonia, et al. “Exosomes: A Source for New and Old Biomarkers in Cancer.” Cancers 2020, Vol. 12, Page 2566, vol. 12, no. 9, Multidisciplinary Digital Publishing Institute, Sept. 2020, p. 2566, doi:10.3390/CANCERS12092566.

Longa, Qianfa, et al. “Intranasal MSC-Derived A1-Exosomes Ease Inflammation, and Prevent Abnormal Neurogenesis and Memory Dysfunction after Status Epilepticus.” Proceedings of the National Academy of Sciences of the United States of America, vol. 114, no. 17, Proc Natl Acad Sci U S A, Apr. 2017, pp. E3536–45, doi:10.1073/PNAS.1703920114.

Monguió-Tortajada, Marta, et al. “Extracellular Vesicle Isolation Methods: Rising Impact of Size-Exclusion Chromatography.” Cellular and Molecular Life Sciences, vol. 76, no. 12, Birkhauser Verlag AG, Mar. 2019, pp. 2369–82, doi:10.1007/S00018-019-03071-Y/METRICS.

Pacienza, Natalia, et al. “In Vitro Macrophage Assay Predicts the In Vivo Anti-Inflammatory Potential of Exosomes from Human Mesenchymal Stromal Cells.” Molecular Therapy - Methods and Clinical Development, vol. 13, no. June, Elsevier Ltd., 2019, pp. 67–76, doi:10.1016/j.omtm.2018.12.003.

Staubach, Simon, et al. “Scaled Preparation of Extracellular Vesicles from Conditioned Media.” Advanced Drug Delivery Reviews, vol. 177, Elsevier, Oct. 2021, p. 113940, doi:10.1016/J.ADDR.2021.113940.

Théry, Clotilde, et al. “Minimal Information for Studies of Extracellular Vesicles 2018 (MISEV2018): A Position Statement of the International Society for Extracellular Vesicles and Update of the MISEV2014 Guidelines.” Journal of Extracellular Vesicles, vol. 7, no. 1, 2018, doi:10.1080/20013078.2018.1535750.

Whiteside, Theresa L. “Tumor-Derived Exosomes and Their Role in Cancer Progression.” Advances in Clinical Chemistry, vol. 74, Adv Clin Chem, 2016, pp. 103–41, doi:10.1016/BS.ACC.2015.12.005.

---

## [Decision Letter · Decision Letter 1]

4 Sep 2023

Ion exchange chromatography as a simple and scalable method to isolate biologically active small extracellular vesicles from conditioned media

PONE-D-23-17535R1

Dear Dr. Malvicini,

We’re pleased to inform you that your manuscript has been judged scientifically suitable for publication and will be formally accepted for publication once it meets all outstanding technical requirements.

Kind regards,

Elia Bari

Academic Editor

PLOS ONE

Additional Editor Comments (optional):

Reviewers' comments:

Reviewer's Responses to Questions

**Comments to the Author**

1. Does the manuscript report a protocol which is of utility to the research community and adds value to the published literature?

Reviewer #2: Yes

2. Has the protocol been described in sufficient detail?

To answer this question, please click the link to protocols.io in the Materials and Methods section of the manuscript (if a link has been provided) or consult the step-by-step protocol in the Supporting Information files.

The step-by-step protocol should contain sufficient detail for another researcher to be able to reproduce all experiments and analyses.

Reviewer #2: Yes

3. Does the protocol describe a validated method?

Reviewer #2: Yes

4. If the manuscript contains new data, have the authors made this data fully available?

Reviewer #2: Yes

**5. Is the article presented in an intelligible fashion and written in standard English?**

Reviewer #2: Yes

6. Review Comments to the Author

Reviewer #2: The manuscript has been improved according to the reviewer's comments. The manuscript might be published in PLOS One in its current form.

Sincerely,

7. PLOS authors have the option to publish the peer review history of their article (what does this mean?). If published, this will include your full peer review and any attached files.

Reviewer #2: **Yes: **Sergey Sedykh

---

## [Editor Report · Acceptance letter]

7 Sep 2023

PONE-D-23-17535R1 

Ion exchange chromatography as a simple and scalable method to isolate biologically active small extracellular vesicles from conditioned media 

Dear Dr. Malvicini:

I'm pleased to inform you that your manuscript has been deemed suitable for publication in PLOS ONE. Congratulations! Your manuscript is now with our production department. 

Kind regards, 

on behalf of

Dr. Elia Bari 

Academic Editor

PLOS ONE